# BAYESIAN DEEP EQUILIBRIUM MODELS WITH SEQUENTIAL INFERENCE

## ABSTRACT

Deep Equilibrium Models (DEQs) have drawn considerable attention due to their unique advantages. However, their uncertainty estimation, crucial for prediction-sensitive applications, remains unexplored. In this paper, we propose Bayesian Deep Equilibrium Models to address this gap for the first time. Our study highlights the substantial computational cost associated with uncertainty estimation in Bayesian DEQs. To mitigate this challenge, we introduce a novel sequential inference approach that captures the similarities in the parameters and reduces computational redundancy in the inference, offering a promising method to accelerate uncertainty quantification in DEQs. We also provide theoretical justification for the motivation behind our approach. Comprehensive experiments on MNIST, CIFAR-10, and ImageNet demonstrate that our method can speed up uncertainty estimation with Bayesian DEQs by up to 3 times without any sacrifice in performance.

## 1 INTRODUCTION

Deep equilibrium models (DEQs) (Bai et al., 2019) have garnered considerable attention in recent years due to their wide applications (Bai et al., 2022; Gu et al., 2020; Bai et al., 2020). Unlike traditional deep neural networks, which define representations through explicit functions, DEQs leverage a fixed-point formulation to implicitly define the underlying representation. Consequently, DEQs can be viewed as weight-tied neural networks with an infinite number of layers. DEQs are recognized for several distinct advantages. First, the implicit nature of their formulation makes them particularly well-suited for solving ordinary differential equations and physical systems (Marwah et al., 2023). This implicit formulation also endows DEQs with unique statistical properties, such as robustness (Liu et al., 2021). Second, DEQs are memory-efficient during training, as gradients are computed using the implicit function theorem instead of storing the computation graph (Bai et al., 2019). Finally, the optimization framework of DEQs offers control over the accuracy-efficiency trade-off, making them highly effective in fast-iterative scenarios (Chen et al., 2018). Collectively, these attributes underscore the growing importance of DEQs in modern deep learning.

Quantifying the uncertainty of models is crucial in real-world applications such as healthcare and autonomous driving, where minor errors can lead to significant consequences (Abdar et al., 2021). This necessitates the development of trustworthy deep models that can indicate the reliability of their predictions. Furthermore, uncertainty estimation is beneficial in tasks such as out-of-distribution detection and active learning (Yang et al., 2024; Settles, 2009). In this context, Bayesian Neural Networks (BNNs) have been proposed as a powerful solution for uncertainty quantification and generalization ability (MacKay, 1992). BNNs model the parameters as a posterior distribution rather than a point estimation, thereby enabling uncertainty estimation. Approaches have been introduced to approximate the posterior, including Monte Carlo Markov Chain (MCMC) (Welling & Teh, 2011), and Stochastic Weight Averaging-Gaussian (SWAG) (Maddox et al., 2019).

A growing body of research has focused on various aspects of DEQs, such as their accuracy, robustness, and efficiency (Bai et al., 2020; Yang et al., 2022; Bai et al., 2021a). However, the uncertainty associated with DEQs remains largely unexplored, raising concerns about the trustworthiness of their application. On the one hand, DEQs are typically trained using stochastic gradient descent (SGD) (Robbins & Monro, 1951), which only provides point estimates for predictions, making it challenging to distinguish aleatoric and epistemic uncertainty (Jospin et al., 2022). On the other

hand, our preliminary study (Figure 1) reveals that DEQs suffer from poor calibration, similar to traditional neural networks (Guo et al., 2017), highlighting the uncertainty estimation issue for DEQs.

In this paper, we present the first comprehensive study on Bayesian Deep Equilibrium Models to address the uncertainty estimation problem in DEQs. Our findings indicate that incorporating MCMC and SWAG into DEQs achieves strong uncertainty estimation. However, it incurs substantial computational costs due to the need for multiple inferences. For instance, MCMC requires at least 30 samples to approximate the posterior distribution, resulting in approximately 80 seconds to estimate the uncertainty for a batch of CIFAR-10 data using MDEQ-LARGE on a single A6000 GPU. This computational burden poses a significant challenge for deploying Bayesian DEQs in sensitive scenarios. In this study, we propose a novel sequential sampling to capture the similarities of the samples in the posterior distribution and further introduce sequential computation to eliminate the redundancy in the inference of DEQs. We integrate these two techniques into a unified framework, termed sequential inference, which significantly accelerates Bayesian DEQs without sacrificing performance. In summary, our contributions are as follows:

- We provide the first study to investigate the uncertainty of DEQs, revealing the inherent challenges. Therefore, we introduce Bayesian DEQs but uncover the computational issue and reveal the redundancy during the inference.

- We propose sequential inference to improve the efficiency of Bayesian DEQs and provide a theoretical justification to clarify the rationale of our motivation.

- We conduct extensive experiments on MNIST, CIFAR-10, and ImageNet to validate the effectiveness of our approach. The results demonstrate that our method can accelerate the inference speed of Bayesian DEQs by up to $3\times$ without sacrificing performance.

## 2 RELATED WORKS

### 2.1 DEEP EQUILIBRIUM MODELS

Recently, there has been a surge of research on deep implicit models defining outputs through implicit functions (Amos & Kolter, 2017; Chen et al., 2018; Bai et al., 2019; Agrawal et al., 2019; El Ghaoui et al., 2021; Bai et al., 2020; Winston & Kolter, 2020). Among these, deep equilibrium models define implicit layers by solving a fixed-point problem (Bai et al., 2019; 2020). With unique advantages, DEQs achieve state-of-the-art performance in various tasks, including image recognition (Bai et al., 2020), image generation (Pokle et al., 2022), graph modeling (Gu et al., 2020; Chen et al., 2022), and solving equations (Marwah et al., 2023). Though DEQs are competitive in terms of performance, the computational inefficiency remains a bottleneck.

### 2.2 BAYESIAN NEURAL NETWORKS

Deep neural networks are often prone to being overconfident (Guo et al., 2017) and provide unreliable uncertainty estimation (Abdar et al., 2021). Bayesian neural networks (BNNs) have been proposed to address this issue by estimating the posterior distribution of the parameters (MacKay, 1992). Monte Carlo Markov Chain (MCMC) constructs Markov chains where the samples generated follow the posterior distribution (Bardenet et al., 2017). Several approaches have adapted MCMC to stochastic gradient descent (Welling & Teh, 2011; Chen et al., 2014). MC Dropout, on the other hand, shows that applying dropout during the inference phase can be interpreted as a Gaussian process, thereby enabling posterior estimation (Gal & Ghahramani, 2016). Stochastic Weight Averaging-Gaussian (SWAG) estimates the posterior using a Gaussian distribution (Maddox et al., 2019) by collecting parameters along the training trajectory of SWA (Izmailov et al., 2018), making it compatible with modern training practices for DNNs. Laplacian Approximation (LA)(Daxberger et al., 2021) approximates the posterior distribution with a Gaussian by leveraging the Hessian matrix. However, such methods either require multiple forward passes during inference, resulting in significant computational overhead, or not suitable for DEQs, as discussed in Section5.3.

Another line of research focuses on uncertainty estimation using a single forward pass, which typically modify the final layer to facilitate uncertainty estimation for efficiency (Liu et al., 2020; Van Amersfoort et al., 2020). However, Bayesian-based methods remain dominant due to their

distinct advantages. First, single-pass methods often yield less reliable uncertainty estimation, particularly on large-scale datasets (Abdar et al., 2021). Second, their deterministic nature makes them prone to overfitting and sensitive to the training and hyperparameters, requiring substantial data to achieve robust performance (Gawlikowski et al., 2023). The limitations underscore the importance of research into ensemble-based uncertainty estimation techniques, which is the focus of this paper.

# 3 Bayesian Deep Equilibrium Models

In this section, we begin with preliminary studies to demonstrate that uncertainty estimation is a challenging task for DEQs. To tackle this issue, we incorporate Bayesian inference methods into DEQs and investigate the computational cost challenges in practical settings.

## 3.1 Deep Equilibrium Models

Explicit neural networks typically perform the forward pass using explicit functions, represented as $\mathbf{z}^{l+1} = f^l(\mathbf{z}^l; \theta_l)$, where $\mathbf{z}^l$ and $\mathbf{z}^{l+1}$ denote the input and output of the layer $f^l(\cdot; \theta_l)$, parameterized by $\theta_l$. In contrast, DEQs define their outputs by a fixed-point problem:

$$\mathbf{z}^* = f(\mathbf{z}^*, \mathbf{x}; \theta), \tag{1}$$

where $\mathbf{z}^*$ is the output representation and $\mathbf{x}$ is the input data. Consequently, DEQs require solving a fixed-point equation for each input $\mathbf{x}$ to compute the representation $\mathbf{z}^*$. Several fixed-point solvers have been employed for DEQs, including fixed-point iteration, Anderson acceleration, and Broyden's method (Bai et al., 2019). For instance, fixed-point iteration updates the solution repeatedly until convergence:

$$\mathbf{z}^{l+1} = f(\mathbf{z}^l, \mathbf{x}; \theta). \tag{2}$$

In this paper, we represent the fixed-point solver as:

$$\mathbf{z} = \text{Solver}(f, \mathbf{x}, \mathbf{z}^0; \theta), \tag{3}$$

where $\mathbf{z}^0$ is the initial guess. Although deep learning achieves high accuracy, its uncertainty estimation remains a significant concern, particularly in sensitive applications. This issue occurs in DEQs as well. First, DEQs trained using SGD only provide point estimates for predictions, limiting their ability to capture uncertainty (Jospin et al., 2022). Moreover, similar to traditional neural networks, DEQs also exhibit overconfidence in their predictions, where overconfidence means the confidence of predictions is higher than the accuracy (Guo et al., 2017). To illustrate this, we evaluate the calibration

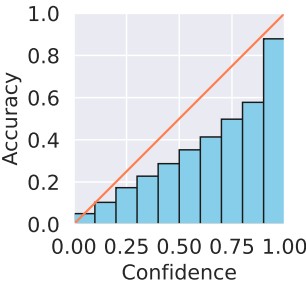

Figure 1: Reliability diagram for MDEQ trained with SGLD on CIFAR-10. The confidence in each bin is higher than the corresponding accuracy.

of MDEQ on CIFAR-10 and plot a reliability diagram comparing the confidence of the model to its accuracy, where MDEQ is a particular implementation of DEQs designed for computer vision tasks, adhering to all the general properties of DEQs. As shown in Figure 1, the confidence in each bin exceeds its accuracy, highlighting a significant overconfidence issue.

## 3.2 Uncertainty Estimation

BNNs (MacKay, 1992) provide a valid solution to address the above challenge. BNNs treat parameters as distributions rather than point estimates, allowing the final predictions to capture uncertainty through statistical information. In this section, we discuss methods for approximating the posterior in large-scale models and analyze the challenges of applying these approaches to DEQs.

**MCMC.** To approximate the posterior distribution, we can employ the Monte Carlo Markov Chain (MCMC) method, which iteratively updates the model distribution through a Markov chain until it converges to the posterior distribution. To facilitate batch-wise training in deep learning, techniques such as Stochastic Gradient Langevin Dynamics (SGLD) have been introduced (Welling & Teh, 2011), which integrate Langevin dynamics with mini-batch gradient updates:

$$\Delta\theta_t = \frac{\epsilon_t}{2}\left(\nabla \log p(\theta_t) + \frac{N}{n}\sum_{i=1}^{n}\nabla \log p(\mathbf{x}_i|\theta_t)\right) + \eta_t, \quad \text{where } \eta_t \sim N(0, \epsilon_t). \tag{4}$$

(a) Sequential Sampling        (b) Sequential Computation

Figure 2: Illustrations of our proposed sequential inference. (a) *Step 1*: sequential sampling draws parameters from the posterior in a sequential manner, capturing parameter similarities. (b) *Step 2*: sequential computation initializes with the previous and converges to the fixed point within a few iterations. Finally, we aggregate the predictions to approximate the output distribution.

Here, $\epsilon$ denotes the learning rate, $p(\cdot)$ represents the likelihood, $n$ is the batch size, and $N$ is the size of the dataset. Compared to SGD, SGLD introduces additional noise at each iteration. By storing the parameters $\theta_t$ in the optimization process, we can approximate the posterior distribution and aggregate the resulting predictions to estimate the expected value.

**SWAG.** Stochastic Weight Averaging-Gaussian (SWAG) approximates the posterior using a Gaussian distribution (Maddox et al., 2019). Suppose we train the model for $T$ epochs. Then the mean of the Gaussian distribution is computed as the average of the parameters over these epochs: $\theta_{\text{SWAG}} = \frac{1}{T} \sum_{i=1}^{T} \theta_i$, where $\theta_i$ denotes the model parameters at epoch $i$. To approximate the covariance of the Gaussian distribution, SWAG defines $\Sigma_{\text{Diag}}$ and $\Sigma_{\text{low-rank}}$:

$$\Sigma_{\text{Diag}} = \text{diag}(\overline{\theta^2} - \theta_{\text{SWAG}}^2), \quad \Sigma_{\text{low-rank}} = \frac{1}{K-1} \sum_{i=T-K+1}^{K} (\theta_i - \overline{\theta}_i)(\theta_i - \overline{\theta}_i)^\top, \quad (5)$$

where $\overline{\theta^2} = \frac{1}{T} \sum_{i=1}^{T} \theta_i^2$ and $\overline{\theta}_i = \frac{1}{i} \sum_{j=1}^{i} \theta_j$. The "diag" operator denotes a diagonal matrix, and $K$ represents the number of weights used to approximate $\Sigma_{\text{low-rank}}$. $\Sigma_{\text{low-rank}}$ captures the correlations between parameters in a memory-efficient manner, while $\Sigma_{\text{Diag}}$ is used to stabilize the covariance matrix. The final covariance matrix is estimated as:

$$\Sigma_{\text{SWAG}} = s(\Sigma_{\text{Diag}} + \Sigma_{\text{low-rank}}), \quad (6)$$

where $s$ is a scaling factor. Note that, in practice, we only store the diagonal elements and $\theta_i - \overline{\theta}$, where $i = T - K + 1, \cdots, T$, instead of directly storing $\Sigma_{\text{SWAG}}$. In this paper, we adapt these BNN methods for DEQs by modifying the training settings, as detailed in Section 5.1. However, this adaptation introduces computational challenges, as demonstrated below.

**Inference of Baysian DEQs is computationally expensive.** There are three primary factors contributing to the high computational cost. First, DEQs often have complex architectures to maintain expressiveness compared to standard models. For example, MDEQs incorporate multiple convolutional layers within each fixed-point iteration. Second, achieving the fixed point requires numerous iterations using a solver. Although some higher-order solvers reduce the number of iterations, they demand more computation time per step. Finally, uncertainty estimation requires running DEQs multiple times to approximate the posterior distribution. For instance, SGLD needs at least 30 samples to approximate the posterior, taking approximately 80 seconds for MDEQ-LARGE to estimate the uncertainty for a batch of CIFAR-10 data on a single A6000 GPU. Collectively, these factors make Bayesian DEQs extremely resource-intensive.

## 4 SEQUENTIAL INFERENCE

Uncertainty estimation typically requires multiple sampling and inferences, resulting in a computational bottleneck. To address this challenge, we propose sequential inference, which is composed of two techniques: sequential sampling and sequential computation. In this section, we first introduce the sequential sampling method, which captures the similarities in the parameters while maintaining the posterior distribution. We then present the sequential computation technique, which exploits the similarities to reduce computational redundancy in the forward process.

### 4.1 SEQUENTIAL SAMPLING

Uncertainty estimation methods require multiple inferences using different parameter samples. However, current approaches for ensembling predictions overlook the similarity among the param-

eters resulting from the posterior distribution. In this section, we propose a sequential sampling method to capture similarities and integrate them into existing Bayesian inference techniques. Our approach aims to "order" the samples such that nearby parameters remain close while preserving their distribution at no additional cost. The proposed sampling strategy is illustrated in Figure 2a.

**MCMC.** We collect parameters from the posterior using a Markov Chain in MCMC. Due to the nature of SGD, consecutive samples tend to be closer to one another. As a result, the samples naturally follow an order in which nearby samples remain close. To validate the feasibility of our method, we conduct a preliminary study on CIFAR-10 using MDEQ trained with SGLD and compute the relative $\ell_2$ distance between the parameters from the first epoch and those from subsequent training epochs. As shown in Figure 3a, the distance between the nearby epochs generally are smaller than others, indicating that our sequential sampling effectively captures the similarity.

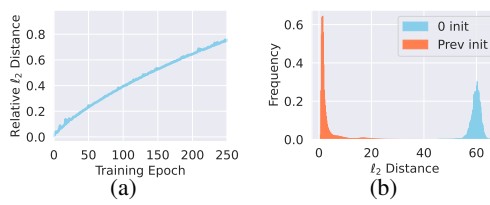

Figure 3: (a) The relative $\ell_2$ distance between the parameters from the first epoch and those from subsequent training epochs in SGLD. (b) The distribution of distances between the initialization and the fixed point with different initializations for SGLD.

**SWAG.** In SWAG, the samples are drawn from the Gaussian distribution $\mathcal{N}(\theta_{\text{SWAG}}, \Sigma_{\text{SWAG}})$. For memory efficiency, we sample parameters from $\mathcal{N}(\theta_{\text{SWAG}}, \Sigma_{\text{SWAG}})$ using the following approach:

$$\hat{\theta} = \theta_{\text{SWAG}} + \sqrt{s}\Sigma_{\text{Diag}}^{\frac{1}{2}}\xi^{(1)} + \frac{\sqrt{s}}{\sqrt{K-1}}\hat{D}\xi^{(2)} \tag{7}$$

where $\hat{D}$ is the concatenation of $\theta_i - \overline{\theta}_i$. Here, $\xi^{(1)} \sim \mathcal{N}(0, I_d)$ corresponds to the dimension of the model parameters, and $\xi^{(2)} \sim \mathcal{N}(0, I_K)$. According to Equation (7), we can utilize Langevin dynamics to sample $\xi^{(1)}$ and $\xi^{(2)}$ from the Gaussian distribution to capture the similarity. We sample $\xi \sim \mathcal{N}(0, I_{d+K})$ and split it into $\xi^{(1)}$ and $\xi^{(2)}$. Considering the probability density function of $\mathcal{N}(0, I_{d+K})$, the samples are drawn in the following sequential manner:

$$\xi_{t+1} = (1-\tau)\xi_t + \sqrt{\tau}\eta, \quad \text{where } \eta \sim \mathcal{N}(0, I_{d+K}) \tag{8}$$

where $\tau$ is the step size of Langevin dynamics.

**Remark 1.** *We clarify that the similarity here is different from auto-correlation in Monte Carlo sampling. Our sequential sampling method "orders" the samples while remaining representative of the posterior distribution. We empirically evaluate the representativeness of samples in sequential sampling in Appendix G, where the Wasserstein-2 distance between the samples from sequential sampling and the normal sampling is pretty small.*

## 4.2 SEQUENTIAL COMPUTATION

In this section, we propose sequential computation to reduce the redundancy of similar forward processes caused by similar parameters captured in sequential sampling. The core idea is to leverage historical information to accelerate subsequent computations. Considering the importance of initialization (Bai et al., 2021a), we utilize the fixed point from the previous iteration as the initialization for the next fixed-point problem, instead of starting from the standard zero initialization. This strategy reduces the initial distance between the starting point and the final fixed-point solution, thereby simplifying the optimization problem. We validate the effectiveness of our initialization by comparing the distance between consecutive fixed points to the distance between a fixed point and zero initialization, following the setting in Section 4.1. As shown in Figure 3b, the previous initialization is significantly closer to the fixed-point solution compared to the standard zero initialization.

To provide an intuitive understanding, consider the fixed-point iteration solver in Equation (3) as analogous to layers in a neural network, where each iteration represents one layer. Using the previous fixed point as the initialization is similar to starting from an intermediate layer, thereby skipping the computations required for the first many layers and eliminating redundancy. For example, if a neural network has 50 layers and our method initializes from an equivalent state at the 45th layer, this approach can achieve up to $10\times$ acceleration. Our method updates the parameters as follows:

$$\mathbf{z}_i^l = \text{Solver}(f, \mathbf{x}, \mathbf{z}_{i-1}^l; \theta_i), \tag{9}$$

where $i = 1, 2, \ldots, N$ and $l$ is the number of the fixed-point iteration to approximate $z_i^*$. We illustrate our method in mini-batch mode on SWAG in Algorithm 1. For other Bayesian methods, please refer to Appendix C. We theoretically demonstrate the feasibility of our approach:

**Theorem 4.1.** *Consider a sequence of parameter samples $\{\theta_1, \theta_2, \ldots, \theta_N\}$ and a DEQ $f(\mathbf{z}, \mathbf{x}; \theta)$. Suppose the following conditions are satisfied:*

- *$f$ has non-sharp fixed points and is smooth within a local region $C \subset \mathbb{R}^d$, where $\{\theta_1, \theta_2, \ldots, \theta_N\} \subset C$. Specifically, there exist constants $M_1 > 0$ and $M_2 > 0$ such that for all $\theta \in C$,*

$$\left\| \left( I - \frac{\partial f}{\partial z}(\cdot; \theta) \right)^{-1} \right\|_2 \leq M_1 \text{ and } \left\| \frac{\partial f}{\partial \theta}(\cdot; \theta) \right\|_2 \leq M_2,$$

- *The parameters in the sequence are sufficiently close to each other: there exists $\delta > 0$ such that for any $0 \leq i \leq N - 1$, $\|\theta_{i+1} - \theta_i\|_2 \leq \delta$.*

*For any $0 \leq i \leq N-1$, the following inequality holds: $\|\mathbf{z}_{i+1}^* - \mathbf{z}_i^*\|_2 \leq \sqrt{d}\delta M_1 M_2$, where $\mathbf{z}^* \in \mathbb{R}^d$ is the solution of fixed-point problems.*

The proof is shown in Appendix A. The above theorem demonstrates that our sequential compatation makes the initialization close to the final fixed point. In practice, a well-trained representation learner will separate features of different classes with a large margin, making most features distant from $\mathbf{0}$. Thus, using the previous fixed point as initialization offers a better starting point compared to a fixed $\mathbf{0}$ initialization. The two assumptions generally hold. First, it is common to assume the smoothness of a well-trained DEQ. Since DEQs converge, the change in fixed points is small, implying that $M_1$, the gradient with respect to $\mathbf{z}$, remains small. Additionally, $M_2$, the gradient with respect to $\theta$, is small, as the model reaches an optimal state during training. Second, the sequential property holds for Bayesian inference methods using our sequential sampling approach. As shown in Appendix I, we empirically approximate the magnitudes of $M_1$ and $M_2$, which supports our assumption.

**Remark 2.** *Our sequential inference is compatible with parallel computing during implementation. First, since multi-model inference typically requires multi-process or multi-node distributed computing (Chandra et al., 2019), which is challenging, it is common to use batch-wise evaluation. This approach allows us to process a batch of data and apply sequential computation in parallel across the batch. Second, even for a single data point, when computational resources (e.g., CPU cores or machines) are limited, we can still accelerate inference by combining sequential inference with parallel computing in a multi-process distributed setup. For example, if we require predictions from 30 models but only have 6 processes available, we execute the ensemble in 5 sequential rounds.*

---

**Algorithm 1** Sequential Inference for SWAG

---

**Require:** DEQ $f(\cdot)$, SWAG mean $\theta_{\text{SWAG}}$, SWAG diagonal $\Sigma_{\text{Diag}}$, SWAG correlation $\hat{D}$, SWAG scaling $s$, Langevin step size $\tau$, input data $\mathbf{x}$
1: Sample Gaussian noise $\xi_0 \in \mathbb{R}^{d+K}$
2: Fixed-point initialization: $\mathbf{z}^0 = \mathbf{0} \in \mathbb{R}^{B \times d}$
3: Prediction posterior initialization: $\mathbf{p}(\mathbf{y}|\mathbf{x}; \theta) = [\ ]$
4: **for** $1 \leq i \leq N$ **do**
5:     Sample Gaussian noise $\eta \in \mathbb{R}^{d+K}$
6:     $\xi_{i+1} = (1 - \tau)\xi_i + \sqrt{\tau}\eta$
7:     $\xi^{(1)} = \xi_{i+1}[0, d], \xi^{(2)} = \xi_{i+1}[d+1, d+K]$
8:     $\theta_i = \theta_{\text{SWAG}} + \sqrt{s}\Sigma_{\text{Diag}}^{1/2}\xi^{(1)} + \frac{\sqrt{s}}{\sqrt{K-1}}\hat{D}\xi^{(2)}$
9:     $\mathbf{z}^i = \text{Solver}(f, \mathbf{x}, \mathbf{z}^{i-1}; \theta_i)$
10:    Classify $\mathbf{z}^i$ to get $\mathbf{p}(\mathbf{y}|\mathbf{x}; \theta_i) \in \mathbb{R}^C$
11:    Append $\mathbf{p}(\mathbf{y}|\mathbf{x}; \theta_i)$ to $\mathbf{p}(\mathbf{y}|\mathbf{x}; \theta)$
12: **end for**
13: **return** $\mathbf{p}(\mathbf{y}|\mathbf{x}; \theta)$

---

## 5 EXPERIMENTS

In this section, we first introduce the experimental settings, including the datasets, model architectures, Bayesian inference methods, and evaluation metrics. Next, we apply these Bayesian inference methods to DEQs and demonstrate how our sequential inference technique integrates with them. Finally, we perform a comprehensive analysis of our sequential inference through ablation studies.

### 5.1 EXPERIMENTAL SETTINGS

**Datasets & DEQ Settings.** We first evaluate the effectiveness of our method on synthetic datasets. Additionally, we use three popular datasets in image recognition, MNIST (LeCun et al., 1998), CIFAR-10 (Krizhevsky et al., 2009), and ImageNet (Deng et al., 2009) to evaluate the scalability of DEQs. We also conduct experiments on UCI-Energy (Tsanas & Xifara, 2012) to show that our method works for real-world regression tasks in Appendix D. Note that our method is general and

can be applied to any DEQ model. For the synthetic dataset and MNIST, we use a DEQ with a fully connected layer (FC-DEQ), following the configuration in MonoDEQ (Winston & Kolter, 2020). For CIFAR-10, we select the Multi-Scale DEQ (MDEQ) with Jacobian regularization (Bai et al., 2020)We evaluate the performance of two variants: MDEQ-SMALL and MDEQ-LARGE, using the settings in the original paper (Bai et al., 2020). For ImageNet, we only evaluate with MDEQ-SMALL. For our sequential inference, we employ a warm-up strategy, where the first batch of fixed points is computed using the iteration threshold to ensure convergence. Detailed information of the model configuration and training strategy is provided in Appendix B.

**Bayesian Methods.** We adopt two Bayesian methods for uncertainty estimation as discussed in Section 3 in the main paper. For MCMC, we choose Stochastic Gradient Langevin Dynamics (SGLD) as the algorithm to sample parameters from the Bayesian posterior. We strictly follow the formulation in the paper (Welling & Teh, 2011) and use the size of the training dataset to define the scaling of the noise term in Langevin dynamics (Welling & Teh, 2011). For SWAG, we approximate the covariance matrix following the paper (Maddox et al., 2019). For the notation clarity, we use the Bayesian inference method (e.g., SGLD, SWAG) appended to the model name to denote the training method. For example, FC-DEQ-SGLD refers to the FC-DEQ model trained with SGLD. For more details on the implementation of posterior estimation in Bayesian models, please refer to Appendix B. To further show the generalization of our methods, we conduct experiments on MC-Dropout (Gal & Ghahramani, 2016), Deep ensembles (Lakshminarayanan et al., 2017), Hamiltonian Monte Carlo (HMC) (Chen et al., 2014), and Bayes by Backpropagation (BBB) (Graves, 2011) in Appendix E.

Note that this is the first work to study uncertainty estimation for DEQ models and we incorporate the SGD training for comparison. Besides, we compare with efficient Linearized Laplacian Approximation (LLA), and Last-Layer Laplacian Approximation (LLLA) (Daxberger et al., 2021).

**Measurement.** As a standard metric for classification tasks, we use the error rate (ERR) to evaluate classification performance. For uncertainty estimation, we adopt negative log-likelihood (NLL) and calibration as evaluation metrics. Calibration measures the alignment between the confidence and accuracy of the predictions. Specifically, we use the expected calibration error (ECE) to assess calibration performance, following Maddox et al. (2019). We compare the measurements using different numbers of function evaluations (**NFE**), which correspond to iterations of fixed points. A smaller NFE indicates higher efficiency. Finally, we provide out-of-distribution results in Appendix F.

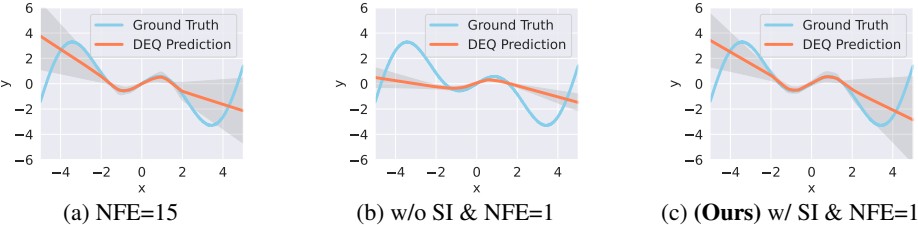

(a) NFE=15          (b) w/o SI & NFE=1          (c) **(Ours)** w/ SI & NFE=1

Figure 4: Uncertainty estimation with FC-DEQ on the toy dataset. **SI** represents our sequential inference. The gray lines correspond to the predicted mean along with three standard deviations.

## 5.2 SYNTHETIC DATASET

In this section, we present results on a toy dataset. The dataset is generated using the function $y = x\cos(x) + \epsilon$, where $\epsilon \sim N(0, 0.01)$. We use data points from the interval $[-2, 2]$ for training and evaluate on $[-5, 5]$. We apply MCMC to train FC-DEQ and conduct inference under three settings: (1) standard inference with 15 fixed-point iterations, (2) standard inference with one fixed-point iteration, and (3) sequential inference with one fixed-point iteration. The results, shown in Figure 4, demonstrate that our method achieves accurate predictions and reliable uncertainty estimation with just one iteration, matching the performance of standard inference with 15 fixed-point iterations.

## 5.3 NUMERICAL EXPERIMENTS

We compare the metrics between standard inference and our sequential inference on MNIST and CIFAR-10. To illustrate the effectiveness, we report the measurements for each NFE up to the maximum number of fixed-point iterations. We use "**w/ SI**" to denote our sequential inference and "**w/o SI**" for standard inference. We use **SGD**, **LLA**, and **LLLA** as baselines for comparison.

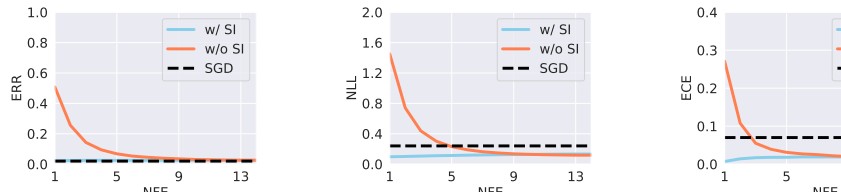

Figure 5: The error rate (ERR), negative log-likelihood (NLL), and expected calibration error (ECE) of FC-DEQ-SGLD on MNIST. **SI** represents our sequential inference.

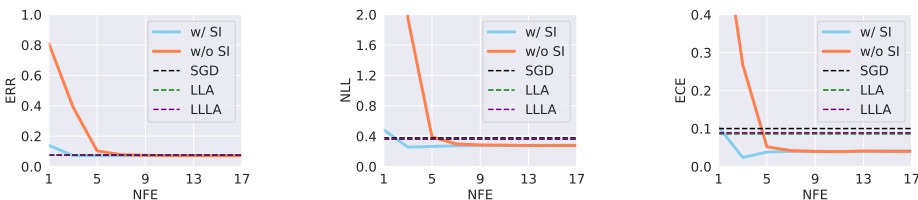

Figure 6: The error rate (ERR), negative log-likelihood (NLL), and expected calibration error (ECE) of MDEQ-LARGE-SGLD on CIFAR-10. **SI** represents sequential inference.

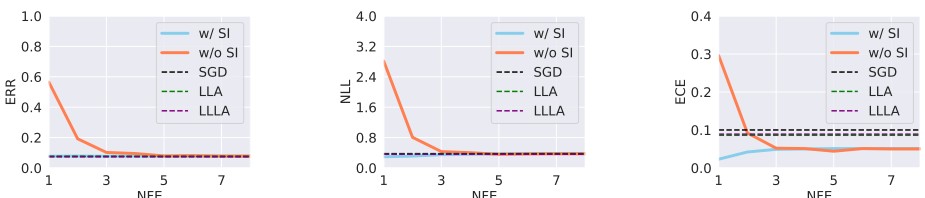

Figure 7: The error rate (ERR), negative log-likelihood (NLL), and expected calibration error (ECE) of MDEQ-LARGE-SWAG on CIFAR-10. **SI** represents sequential inference.

We present the results on CIFAR-10 using MCMC and SWAG with MDEQ-LARGE in Figures 6 and 7, respectively. To demonstrate that our sequential inference is generalizable across different datasets and architectures, we show the results of FC-DEQ-SGLD on MNIST in Figures 5. Additionally, we present the running time of the experiments using SGLD in Figure 8, where all timings are recorded on a single A6000 GPU. Due to the page limitation, more results across all architectures and ImageNet are provided in Appendix D. We can draw the following observations.

**Faster Convergence.** Our proposed sequential inference significantly accelerates uncertainty estimation for Bayesian DEQs. The error rate, negative log-likelihood, and expected calibration error decrease more rapidly with our sequential inference on both MNIST and CIFAR-10. Sequential inference consistently outperforms standard inference with the same number of iterations. For example, in Figure 5, sequential inference requires only 1 step to achieve the same error rate (2.6%) as the maximum number of iterations for all methods, whereas standard inference requires at least 6 steps to reach a higher error rate (5.3%). On CIFAR-10, sequential inference needs only 2 steps (7.2%) to match the performance of standard inference at 9 steps (7.4%). From a runtime perspective, our method achieves substantial speedups across all architectures. For MDEQ-SMALL-SGLD, sequential inference requires 69 seconds to reach a low error rate (16.7%), compared to 209 seconds with standard inference (16.5%), resulting in a $3\times$ acceleration.

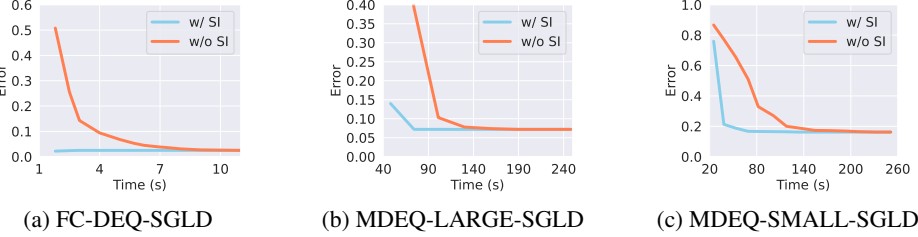

(a) FC-DEQ-SGLD  (b) MDEQ-LARGE-SGLD  (c) MDEQ-SMALL-SGLD

Figure 8: The error rate (Error) with respect to the running time of SGLD.

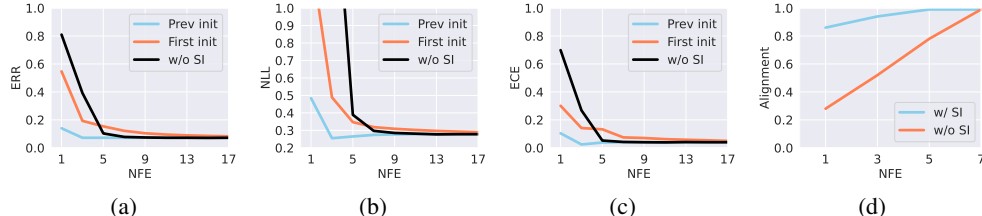

(a)          (b)          (c)          (d)

Figure 9: Ablation studies for sequential inference. Comparison of the (a) error rate (ERR), (b) negative log-likelihood (NLL), and (c) expected calibration error (ECE) between the previous fixed point and the first fixed point. (d) The ratio that the predictions are consistent with the ones with the max number of evaluations.

**Maintaining Performance.** Our method not only accelerates inference but also maintains strong performance across evaluation metrics. All the Bayesian methods provide stronger performance compared to SGD, addressing the unreliable problem. For instance, the ECE of SGD achieves $9\%$ while SWAG only has $0.5\%$ on CIFAR-10. Besides, the results indicate that Laplacian Approximation offers no significant improvement over standard SGD. The suboptimal results may stem from the unrolled computation scheme and the instability of the Jacobian matrices, both of which can hinder effective approximation (Bai et al., 2021b). Moreover, the uncertainty estimation is slightly improved when NFE is small in Figure 5. This trend is similar to the findings of Guo et al. (2017), where the negative log-likelihood increases as the training epoch increases. We argue that this behavior is a result of the accuracy-efficiency trade-off inherent in DEQs, which avoids overconfidence.

### 5.4 Ablation Study

To better understand how our sequential inference works, we conduct comprehensive ablation studies by evaluating the effectiveness of our sequential sampling and analyzing the instance-level performance. Additionally, we investigate the impact of the warm-up strategy in Appendix H and show ablation studies on the hyperparameters of the uncertainty estimation methods and Langevin Sampling in Appendix J. Finally, we provide memory usage study in Appendix K.

**Effectiveness of Sequential Sampling.** To show the effectiveness of sequential sampling, we also validate whether the previous fixed point is a better initialization compared to other options. We conduct experiments using the first fixed point in the sequence as the initialization for comparison with respect to the measurements. The results for MDEQ-LARGE-SGLD are shown in Figure 9. The previous fixed point consistently outperforms the first one as an initialization, showing the effectiveness of similar parameters. In addition, we present an ablation study on the Langevin sampling step size $\tau$ used in SWAG, detailed in Appendix J.

**Instance-Level Test.** In the main experiments, we evaluate the performance of our method on both prediction and uncertainty estimation. However, global performance alone is insufficient to fully demonstrate the effectiveness of our approach. Our method modifies the initialization, which could potentially alter the final predictions. To show that our method can closely align with standard DEQ inference, we perform an instance-level ablation study. We measure the percentage of labels that match those obtained by standard DEQ inference that have reached the fixed point. As illustrated in Figure 9d, our method quickly aligns with the standard DEQ inference, achieving an alignment rate close to $1$. In contrast, without sequential inference, DEQs with the same number of evaluations exhibit slower alignment of the final fixed points, requiring more than 6 steps.

### 6 Conclusion

This paper presents the first study on uncertainty estimation for DEQs, demonstrating that uncertainty estimation can pose a significant challenge for these models. To address this, we propose Bayesian DEQs and identify that the associated computational cost is substantial. We further introduce sequential sampling to capture similarities in uncertainty estimation and sequential computation to accelerate the process by mitigating the redundancy during the inference. Comprehensive experiments show that our method can achieve up to a $3\times$ speedup without any performance degradation on Bayesian DEQs. One limitation of our approach is that certain uncertainty estimation methods, such as deep ensembles, are difficult to directly apply our sequential sampling to capture the similarities, which slightly restricts the generalization of our method. Nevertheless, our findings and approach provide valuable insights and represent a significant step toward the trustworthy deployment of DEQs, potentially advancing the security of real-world AI applications.

## 7 ETHICS STATEMENT

Our paper studies Bayesian Deep Equilibrium Models, and proposes sequential inference to accelerate the inference speed. To the best of our knowledge, our work does not involve human subjects, practices to data set releases, potentially harmful insights, methodologies and applications, potential conflicts of interest and sponsorship, discrimination/bias/fairness concerns, privacy and security issues, legal compliance, and research integrity issues.

## 8 REPRODUCIBILITY STATEMENT

Our paper includes all the necessary resources to reproduce our method. Detailed implementation settings are provided in Section 5.1 and Appendix B. We will release the code after the paper is accepted.

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

## A  PROOF FOR THEOREM 4.1

**Theorem A.1** (Restated, 4.1). *Consider a sequence of parameter samples $\{\theta_1, \theta_2, \ldots, \theta_N\}$ and a DEQ $f(\mathbf{z}, \mathbf{x}; \theta)$. Suppose the following conditions are satisfied:*

- *$f$ has non-sharp fixed points and is smooth within a local region $C \subset \mathbb{R}^d$, where $\{\theta_1, \theta_2, \ldots, \theta_N\} \subset C$. Specifically, there exist constants $M_1 > 0$ and $M_2 > 0$ such that for all $\theta \in C$,*

$$\left\| \left( I - \frac{\partial f}{\partial z}(\cdot; \theta) \right)^{-1} \right\|_2 \leq M_1 \quad and \quad \left\| \frac{\partial f}{\partial \theta}(\cdot; \theta) \right\|_2 \leq M_2, \tag{10}$$

- *The parameters in the sequence are sufficiently close to each other: there exists $\delta > 0$ such that for any $0 \leq i \leq N - 1$,*

$$\|\theta_{i+1} - \theta_i\|_2 \leq \delta. \tag{11}$$

*For any $0 \leq i \leq N - 1$, the following inequality holds:*

$$\|\mathbf{z}_{i+1}^* - \mathbf{z}_i^*\|_2 \leq \sqrt{d}\delta M_1 M_2,$$

*where $\mathbf{z}^* \in \mathbb{R}^d$ is the solution of fixed-point problems.*

**Proof:** According to the implicit function theorem, the equation $z = f(z, x; \theta)$ has an implicit function denoted by $\varepsilon(x; \theta)$. Define $\Delta\theta_i := \theta_{i+1} - \theta_i$, then we have:

$$\|z_{i+1}^* - z_i^*\|_2^2 = \|\varepsilon(x; \theta_{i+1}) - \varepsilon(x; \theta_i)\|_2^2$$
$$= \|\varepsilon(x; \theta_i + \Delta\theta_i) - \varepsilon(x; \theta_i)\|_2^2.$$

By the Mean Value Theorem, there exists a series of $\theta_i^j$ such that:

$$\|\varepsilon(x; \theta_i + \Delta\theta_i) - \varepsilon(x; \theta_i)\|_2^2 = \sum_{j=1}^{d} \|\varepsilon_j(x; \theta_i + \Delta\theta_i) - \varepsilon_j(x; \theta_i)\|_2^2$$

$$= \sum_{j=1}^{d} \|\varepsilon_j(x; \theta_i) + \nabla_\theta \varepsilon_j(x; \theta_i^j)^T \Delta\theta_i - \varepsilon_j(x; \theta_i)\|_2^2$$

$$= \sum_{j=1}^{d} \|\nabla_\theta \varepsilon_j(x; \theta_i^j)^T \Delta\theta_i\|_2^2,$$

where $\varepsilon_j$ represents the $j$-th component of $\varepsilon$. Now, using our assumptions and applying the implicit function theorem again, we get:

$$\|z_{i+1}^* - z_i^*\|_2^2 = \sum_{j=1}^{d} \|\nabla_\theta \varepsilon_j(x; \theta_i^j)^T \Delta\theta_i\|_2^2$$

$$\leq \sum_{j=1}^{d} \left\| \left( I - \frac{\partial f}{\partial z}(z^*, x; \theta_i^j) \right)^{-1} \right\|_2^2 \left\| \frac{\partial f}{\partial \theta}(z^*, x; \theta_i^j) \right\|_2^2 \|\Delta\theta_i\|_2^2$$

$$\leq d\delta^2 M_1^2 M_2^2,$$

where $z^*$ is the fixed point corresponding to the given inputs and parameters. $\qquad\square$

# B  EXPERIMENT DETAILS

## B.1  MODEL ARCHITECTURE

**FC-DEQ.** We use fully connected deep equilibrium models (FC-DEQ) for the classification task on the toy dataset and MNIST, following (Winston & Kolter, 2020). Specifically, we define the fixed-point problem using the following formula:

$$z = \sigma(U_l x + V_l \sigma(U_{l-1} x + V_{l-1} \cdots \sigma(U_1 x + V_1 z + b_1) + b_{l-1}) + b_l), \tag{12}$$

where $U_i$, $V_i$ are weight matrices, $b_i$ are bias terms, and $\sigma$ is the activation function. For our toy regression task, we set the number of layers to 2 and the dimension of the hidden embedding to 32. For MNIST, we set the number of layers to 2 and the dimension of the hidden embedding to 1000. The activation function $\sigma$ of both experiments is chosen as ReLU.

**MDEQ.** Multi-resolution deep equilibrium models (MDEQ) are a class of implicit networks well-suited to large-scale and highly hierarchical pattern recognition tasks. They are inspired by modern computer vision deep neural networks, which leverage multi-resolution techniques to learn features at different scales. These simultaneously learned multi-resolution features allow us to train a single model on a diverse set of tasks and loss functions. For example, a single MDEQ model can be trained to perform both image classification and semantic segmentation. MDEQs have been shown to match or exceed the performance of recent competitive computer vision models, achieving high accuracy in sequence modeling and other tasks.

We report the model hyperparameters in Table 1. For CIFAR-10, we employ both MDEQ-SMALL and MDEQ-LARGE architectures, while for ImageNet, we utilize only MDEQ-SMALL. Most of the hyperparameters remain consistent with the settings specified in the work of (Bai et al., 2021b). MDEQs define several resolutions in the implicit layer, aligning with the design principles of ResNet in computer vision models. The main difference between the architectures lies in the channel size and resolution level. Additionally, all models incorporate GroupNorm, as outlined in the standard MDEQ model from (Bai et al., 2020).

## B.2  TRAINING SETTING

We report the training hyperparameters in Table 2. Following work (Bai et al., 2021b), we use Jacobian regularization to ensure stability during the training process. Moreover, to prevent overfit-

| | CIFAR-10 | | ImageNet |
| | SMALL | LARGE | SMALL |
| --- | --- | --- | --- |
| Input Size | $32 \times 32$ | $32 \times 32$ | $224 \times 224$ |
| Block | BASIC | BASIC | BOTTLENECK |
| Number of Branches | 3 | 4 | 4 |
| Number of Channels | [8, 16, 32] | [32, 64, 128, 256] | [32, 64, 128, 256] |
| Number of Head Channels | [7, 14, 28] | [14, 28, 56, 112] | [28, 56, 112, 224] |
| Final Channel Size | 200 | 1680 | 2048 |

Table 1: Model hyperparameters of MDEQ in our experiments.

| | MNIST | CIFAR-10 | |
| | FC-DEQ | SMALL | LARGE |
| --- | --- | --- | --- |
| Batch Size | 512 | 96 | 96 |
| Epochs | 450 | 120 | 220 |
| Learning Rate | 0.01 | 0.001 | 0.001 |
| Learning Rate Schedule | Linear | Linear | Linear |
| Momentum | 0.0 | 0.98 | 0.98 |
| Jacobian Reg. Strength | 0 | 0.5 | 0.4 |
| Jacobian Reg. Frequency | 0 | 0.05 | 0.02 |

Table 2: Training hyperparameters in our experiments.

ting, we employ data augmentation techniques such as random cropping and horizontal flipping for CIFAR-10, which are commonly utilized in various computer vision tasks.

### B.3 BAYESIAN INFERENCE SETTING

In this section, we present the details of Bayesian inference. For MCMC, we choose Stochastic Gradient Langevin Dynamics (SGLD) as the algorithm to sample parameters from the Bayesian posterior. We strictly follow the formulation in the paper (Welling & Teh, 2011) and use the size of the training dataset to define the scaling of the noise term in Langevin dynamics (Welling & Teh, 2011). The learning rate is selected from $\{0.5, 0.1, 0.01, 0.001\}$ for both FC-DEQ and MDEQ. We set the sampling frequency to 2. For the prior distribution, we use Gaussian distribution, which is identical to weight decay. We set the weight decay as $5 \times 10^{-4}$ for both MNIST and CIFAR-10. For MC Dropout, we apply variational dropout as recommended in the MDEQ training recipe (Bai et al., 2020), selecting the dropout rate from $\{0.01, 0.05, 0.1, 0.3, 0.5\}$. For SWAG, we approximate the covariance matrix with a scaling factor of 0.5, following the paper (Maddox et al., 2019). We sample 100 models for FC-DEQs and 30 models for MDEQs for both MC Dropout and SWAG.

For the implementation of the Laplace approximation, we closely follow the code outlined in Laplace Redux (Daxberger et al., 2021). We adopt both the linearized Laplace approximation and the last-layer Laplace approximation as baselines to evaluate the efficiency of uncertainty estimation.

## C  ALGORITHMS

In this section, we present the algorithms for MCMC and MC Dropout in Algorithm 2 and 3.

## D  RESULTS ON OTHER ARCHITECTURES AND DATASETS

In this section, we present additional results for FC-DEQs and MDEQ-SMALL, illustrated in Figures 10, 11, and 12. The results demonstrate that our method can significantly accelerate uncertainty estimation across different DEQs. Specifically, the error rate decreases rapidly with our sequential inference method, requiring only a single step to achieve strong performance across all metrics. In contrast, other methods take considerably longer: MDEQ-SMALL-SGLD requires at least 9 steps,

---

**Algorithm 2** Sequential Inference for SGLD

---

**Require:** DEQ $f(\cdot)$, training epoch $T$, burn-in $m$, sampling frequency $s$, input data $\mathbf{x}$
1: Parameter samples initialization: $\Theta = [\ ]$
2: **for** $1 \leq i \leq T$ **do**
3:     Train $f$ for one epoch and get parameters $\theta_i$
4:     **if** $i \geq m$ and $mod(i, s) = 0$ **then**
5:         Append $\theta_i$ to $\Theta$
6:     **end if**
7: **end for**
8: Fixed-point initialization: $\mathbf{z}^0 = \mathbf{0} \in \mathbb{R}^{B \times d}$
9: Prediction posterior initialization: $\mathbf{p}(\mathbf{y}|\mathbf{x}; \theta) = [\ ]$
10: **for** $1 \leq i \leq \lfloor (T-m)/s \rfloor$ **do**
11:     $\mathbf{z}^i = \text{Solver}(f, \mathbf{x}, \mathbf{z}^{i-1}; \theta_i)$
12:     Classify $\mathbf{z}^i$ to get $\mathbf{p}(\mathbf{y}|\mathbf{x}; \theta_i) \in \mathbb{R}^C$
13:     Append $\mathbf{p}(\mathbf{y}|\mathbf{x}; \theta_i)$ to $\mathbf{p}(\mathbf{y}|\mathbf{x}; \theta)$
14: **end for**
**Return:** $\mathbf{p}(\mathbf{y}|\mathbf{x}; \theta)$

---

**Algorithm 3** Sequential Inference for MC Dropout

---

**Require:** DEQ $f(\cdot)$, dropout rate $p$, number of inference $N$, input data $\mathbf{x}$
1: Fixed-point initialization: $\mathbf{z}^0 = \mathbf{0} \in \mathbb{R}^{B \times d}$
2: Prediction posterior initialization: $\mathbf{p}(\mathbf{y}|\mathbf{x}; \theta) = [\ ]$
3: **for** $1 \leq i \leq N$ **do**
4:     Apply dropout mask with rate $p$ to $f$
5:     $\mathbf{z}^i = \text{Solver}(f, \mathbf{x}, \mathbf{z}^{i-1}; \theta_i)$
6:     Classify $\mathbf{z}^i$ to get $\mathbf{p}(\mathbf{y}|\mathbf{x}; \theta_i) \in \mathbb{R}^C$
7:     Append $\mathbf{p}(\mathbf{y}|\mathbf{x}; \theta_i)$ to $\mathbf{p}(\mathbf{y}|\mathbf{x}; \theta)$
8: **end for**
**Return:** $\mathbf{p}(\mathbf{y}|\mathbf{x}; \theta)$

---

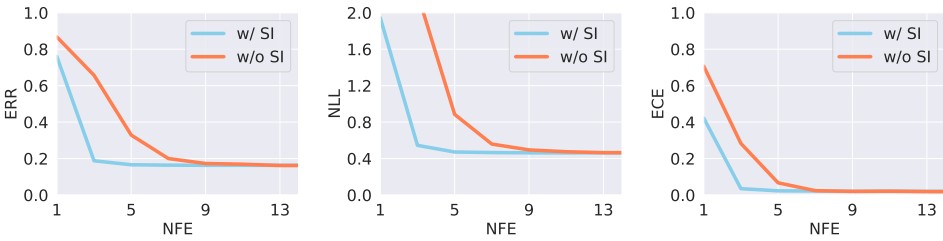

Figure 10: The error rate (ERR), negative log-likelihood (NLL), and expected calibration error (ECE) of MDEQ-SMALL-SGLD on CIFAR-10. **SI** represents our sequential inference.

FC-DEQ-SWAG requires 9 steps, and MDEQ-SMALL-SWAG requires 4 steps. This consistent trend highlights the generalization capability of our approach.

Figure 13 presents the results on ImageNet using MDEQ-SMALL, which align with those observed on smaller datasets.

**UCI Energy.** To further demonstrate the effectiveness of sequential inference, we evaluate performance on the UCI-Energy dataset by computing the log-likelihood of SGLD (higher is better). As shown in the Table 3, our method consistently achieves a faster convergence rate and improved efficiency on this task.

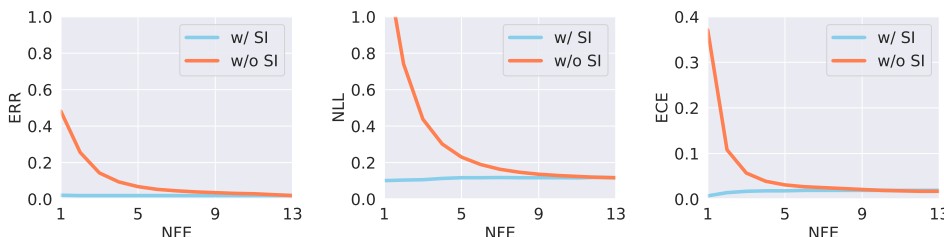

Figure 11: The error rate (ERR), negative log-likelihood (NLL), and expected calibration error (ECE) of FC-DEQ-SWAG on MNIST. **SI** represents our sequential inference.

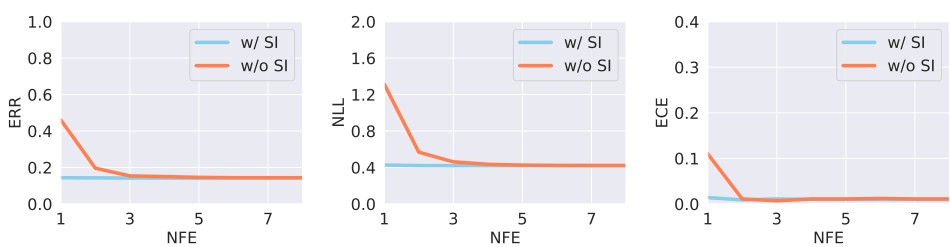

Figure 12: The error rate (ERR), negative log-likelihood (NLL), and expected calibration error (ECE) of MDEQ-SMALL-SWAG on CIFAR-10. **SI** represents our sequential inference.

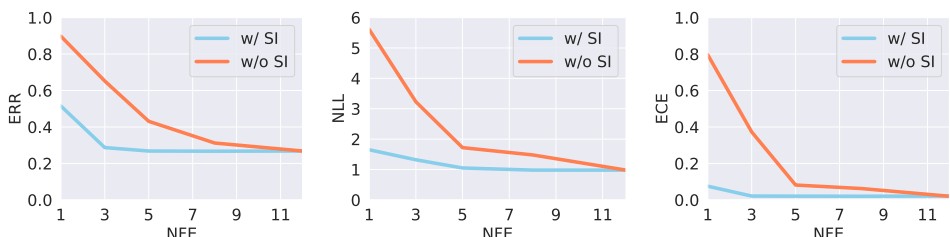

Figure 13: The error rate (ERR), negative log-likelihood (NLL), and expected calibration error (ECE) of MDEQ-SMALL-SWAG on ImageNet. **SI** represents our sequential inference.

## E  RESULTS ON OTHER BASELINES

In this section, we present the performance of our sequential inference on other baselines.

**Monte Carlo Dropout.** Previous studies have shown that modern dropout techniques in deep neural networks can be interpreted as a Gaussian process, providing a theoretical basis for uncertainty estimation (Gal & Ghahramani, 2016). The training is identical to SGD with dropout, but dropout is applied during the inference phase as well. By ensembling predictions across different dropout masks, we can approximate the posterior distribution of the model.

It is not straightforward to perform sequential sampling with MC Dropout, as the algorithm already defines its sampling procedure. However, the typically low dropout rate leads to similar outputs. We approximately treat these outputs as our sequential samples. The results in Appendix D demonstrate that the standard MC Dropout setting is sufficient to capture the redundancy in the outputs.

We report results for MC Dropout in Figures 14, 15, and 16. These results show that our method can also accelerate MC Dropout, even though it does not inherently involve sequential sampling, as dis-

| NFE | 1 | 3 | 5 | 8 | 12 |
|---|---|---|---|---|---|
| SGLD | -19.75 | -1.48 | -1.07 | -1.02 | -1.01 |
| SGLD+SI | -1.06 | -1.01 | -1.01 | -1.01 | -1.01 |

Table 3: The log likelihood of SGLD using sequential inference (SI) with varying numbers of function evaluations (NFE) on UCI dataset.

cussed in the main paper. For example, our sequential inference method reaches strong performance in just 1 step, whereas MC Dropout requires at least 5 steps to achieve comparable results.

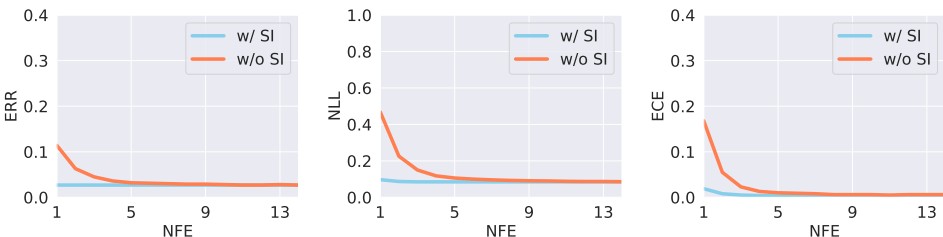

Figure 14: The error rate (ERR), negative log-likelihood (NLL), and expected calibration error (ECE) of FC-DEQ-DROP on CIFAR-10. **SI** represents our sequential inference.

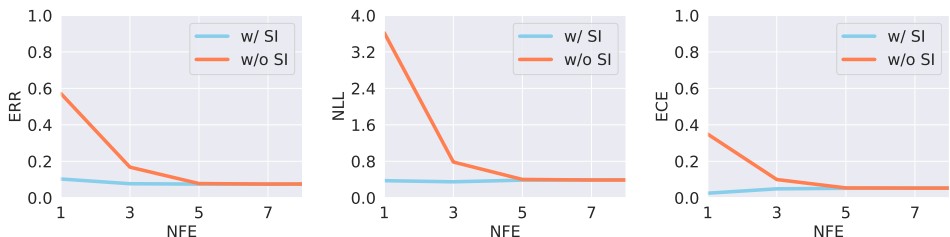

Figure 15: The error rate (ERR), negative log-likelihood (NLL), and expected calibration error (ECE) of MDEQ-LARGE-DROP on CIFAR-10. **SI** represents our sequential inference.

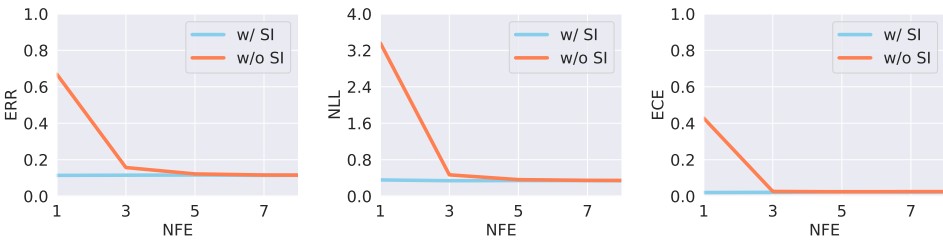

Figure 16: The error rate (ERR), negative log-likelihood (NLL), and expected calibration error (ECE) of MDEQ-SMALL-DROP on CIFAR-10. **SI** represents our sequential inference.

**Deep Ensembles.** Our method also works for deep ensembles. Since deep ensembles train several models with different initializations, sequential sampling is not applicable to them. We directly apply sequential computation to those pre-trained weights. The results are shown in Table 4. While its acceleration performance is not as strong as that of other Bayesian methods (as discussed in the limitations), it still achieves faster inference than the standard approach.

| NFE | 1 | 3 | 5 | 8 | 12 |
|---|---|---|---|---|---|
| Error (Deep Ensemble) | 28.2% | 14.3% | 4.6% | 1.8% | 1.8% |
| Error (Deep Ensemble+SI) | 5.4% | 1.9% | 1.8% | 1.8% | 1.8% |
| ECE (Deep Ensemble) | 18.3% | 6.5% | 2.0% | 1.5% | 1.5% |
| ECE (Deep Ensemble+SI) | 10.7% | 2.2% | 1.6% | 1.5% | 1.4% |

Table 4: The empirical calibration error (ECE) and error rate of Deep Ensemble using sequential inference (SI) with varying numbers of function evaluations (NFE). We report the results using FC-DEQ on MNIST.

**Hamiltonian Monte Carlo.** In addition to SGLD, we conduct experiments on HMC to show that our method works for a faster convergence algorithm, HMC. As shown in Table 5, our method can consistently accelerate the convergence, and the benefit of our method is not limited to SGLD.

| NFE | 1 | 3 | 5 | 8 | 12 |
|---|---|---|---|---|---|
| Error (HMC) | 31.2% | 12.3% | 5.8% | 3.1% | 2.2% |
| Error (HMC+SI) | 4.4% | 2.2% | 2.2% | 2.2% | 2.2% |
| ECE (HMC) | 19.3% | 4.5% | 2.6% | 1.5% | 1.5% |
| ECE (HMC+SI) | 2.7% | 1.5% | 1.5% | 1.5% | 1.5% |

Table 5: The empirical calibration error (ECE) and error rate of HMC using sequential inference (SI) with varying numbers of function evaluations (NFE). We report the results using FC-DEQ on MNIST.

**Bayes by Backpropagation.** To show that our sequential inference works on variational inference methods, we conduct experiments with BBB. As shown in Table 6, our method can consistently improve the convergence speed based on BBB.

| NFE | 1 | 3 | 5 | 8 | 12 |
|---|---|---|---|---|---|
| Error (BBB) | 50.6% | 14.3% | 6.8% | 4.5% | 2.9% |
| Error (BBB+SI) | 2.4% | 2.2% | 2.3% | 2.3% | 2.4% |
| ECE (BBB) | 27.1% | 5.5% | 3.1% | 2.2% | 1.9% |
| ECE (BBB+SI) | 1.1% | 1.8% | 1.9% | 1.9% | 1.8% |

Table 6: The empirical calibration error (ECE) and error rate of BBB using sequential inference (SI) with varying numbers of function evaluations (NFE). We report the results using FC-DEQ on MNIST.

## F    RESULTS ON OUT-OF-DISTRIBUTION DETECTION

Out-of-distribution (OOD) detection is often viewed as a measure of uncertainty estimation. A model with robust uncertainty estimation can effectively detect whether input data is out of distribution. In this study, we report the entropy values for different models, where higher entropy indicates better performance. We utilize Fashion-MNIST (Xiao et al., 2017) as the OOD dataset for MNIST, and CIFAR-100 (Krizhevsky et al., 2009) as the OOD dataset for CIFAR-10. As shown in Table 7, our method (w/ SI) consistently outperforms the standard inference approach. For each comparison, we set the number of inferences to the maximum value.

## G    THE EFFECTIVENESS OF SEQUENTIAL SAMPLING

In the main paper, we use sequential sampling to sample the parameters in SWAG. To show that we do not change the representivity of the samples, we compute the Wasserstein-2 distance between the samples (the number of samples is 30), which is only 0.0013. It means that the samples from our sequential sampling can be treated as the standard sampling considering the representative property.

| Models | Inference Mode | Entropy |
|---|---|---|
| FC-DEQ-SGLD | w/ SI | 0.194 |
| FC-DEQ-SGLD | w/o SI | 0.174 |
| MDEQ-LARGE-SGLD | w/ SI | 0.402 |
| MDEQ-LARGE-SGLD | w/o SI | 0.395 |
| MDEQ-SMALL-SGLD | w/ SI | 0.346 |
| MDEQ-SMALL-SGLD | w/o SI | 0.280 |

Table 7: Entropy for Out-of-Distribution Data with SGLD

## H  THE EFFECTIVENESS WARM-UP STRATEGY

In the main results, we employ a warm-up strategy to provide a good starting point for sequential inference. Here, we evaluate the impact of the warm-up strategy. We conduct experiments on CIFAR-10 using MDEQ-LARGE-SGLD and compare the performance of three configurations: with warm-up, without warm-up, and without sequential inference. As shown in Figure 17, the warm-up strategy significantly enhances performance compared to the setting without warm-up, particularly at the initial stages. Moreover, we observe that even without warm-up, our sequential inference consistently outperforms the standard inference.

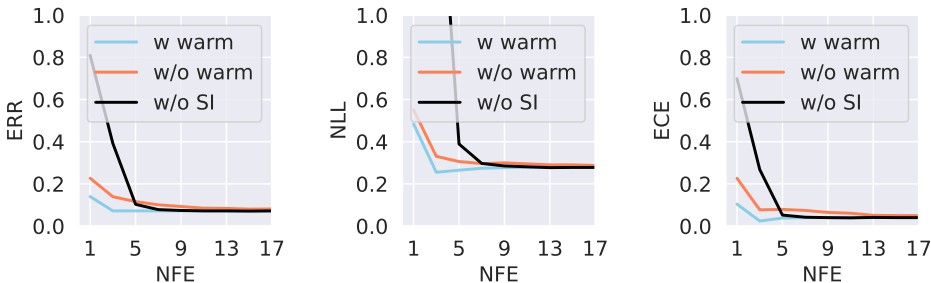

Figure 17: The error rate (ERR), negative log-likelihood (NLL), and expected calibration error (ECE) of MDEQ-LARGE-SGLD on CIFAR-10 for warm-up ablation study. "w/o SI" represents the standard DEQ inference with the same number of evaluations.

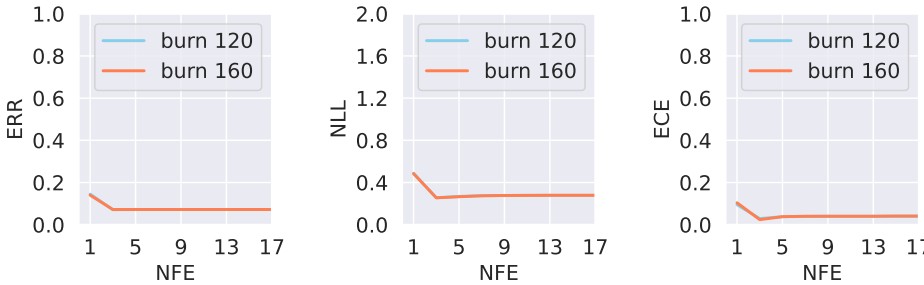

Figure 18: The error rate (ERR), negative log-likelihood (NLL), and expected calibration error (ECE) of MDEQ-LARGE-SGLD on CIFAR-10 for the ablation study of burn-in. "w/o SI" represents the standard DEQ inference with the same number of evaluations.

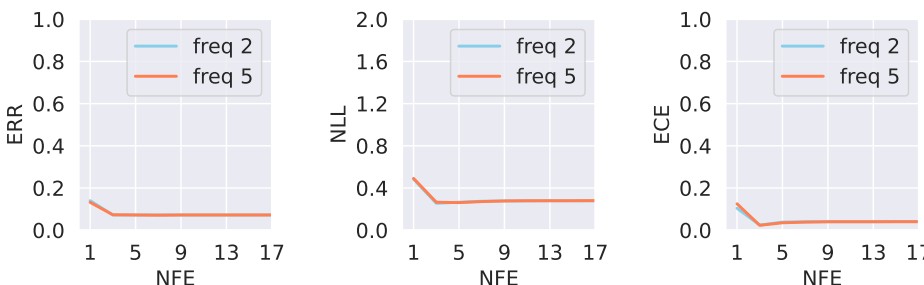

Figure 19: The error rate (ERR), negative log-likelihood (NLL), and expected calibration error (ECE) of MDEQ-LARGE-SGLD on CIFAR-10 for the ablation study of sampling frequency. "w/o SI" represents the standard DEQ inference with the same number of evaluations.

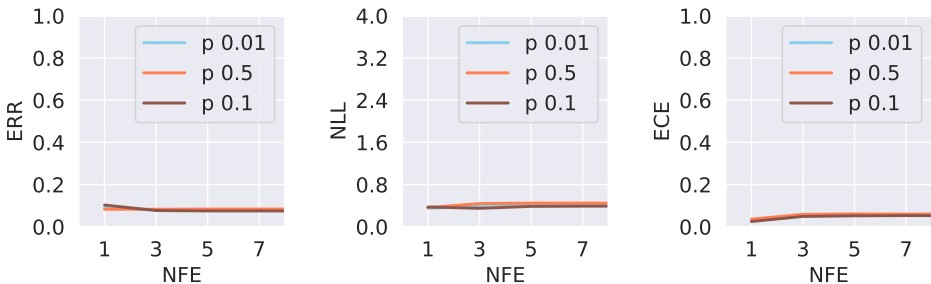

Figure 20: The error rate (ERR), negative log-likelihood (NLL), and expected calibration error (ECE) of MDEQ-LARGE-DROP on CIFAR-10 for the ablation study of dropout rate. "w/o SI" represents the standard DEQ inference with the same number of evaluations.

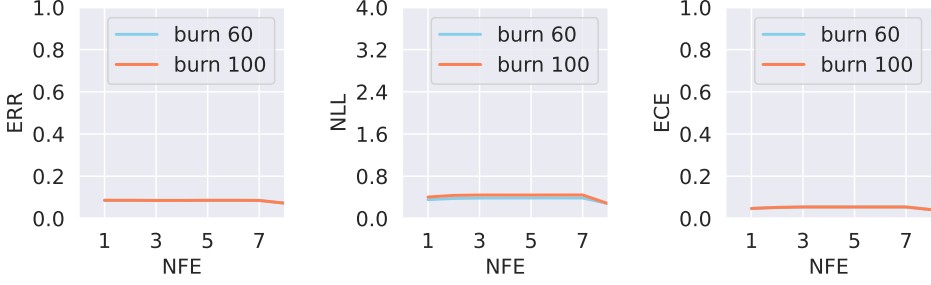

Figure 21: The error rate (ERR), negative log-likelihood (NLL), and expected calibration error (ECE) of MDEQ-LARGE-SWAG on CIFAR-10 for the ablation study of burn-in. "w/o SI" represents the standard DEQ inference with the same number of evaluations.

## I EMPIRICAL STUDIES ON THE ASSUMPTION OF THE THEORY

In Theorem 4.1, the first assumption is about the property of DEQs. To verify our assumption empirically, we estimate the $M_1$ and $M_2$ for MDEQ-SMALL trained by SGLD on CIFAR-10 (for the memory limitation, we only use the small one for illustration). We compute the maximum norm in the assumption over 1000 test samples to estimate $M_1$ and $M_2$. The estimated $M_1$ is 1 while $M_2$ is $1.6 \times 10^{-3}$. To examine a larger-scale scenario, we evaluate MCMC-SMALL trained with

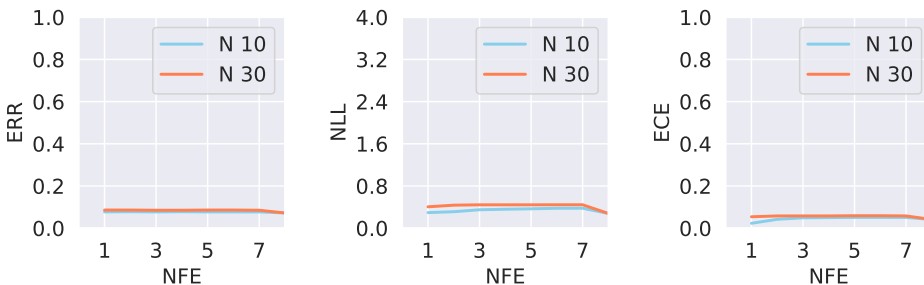

Figure 22: The error rate (ERR), negative log-likelihood (NLL), and expected calibration error (ECE) of MDEQ-LARGE-SWAG on CIFAR-10 for the ablation study of the number of sampling $N$. "w/o SI" represents the standard DEQ inference with the same number of evaluations.

SWAG on ImageNet. Using 100 test samples, we obtain $M_1 = 1.002$ and $M_2 = 4.1 \times 10^{-3}$. These empirical findings suggest that the assumption made in Theorem 4.1 is reasonable in practice.

## J   ABLATION STUDIES

**Bayesian Methods**. In this section, we explore how the hyperparameters of MCMC, MC Dropout, and SWAG influence the performance of our method. As illustrated in Figures 18 and 19, we investigate how burn-in and sampling frequency impact the performance in Stochastic Gradient Langevin Dynamics (SGLD). Specifically, we set the burn-in periods to 120 and 160, and the sampling frequencies to 2 and 5. In addition, we evaluate the sensitivity of the prior in SGLD in Table 8. For MC Dropout, we analyze the effect of the dropout rate on performance, selecting dropout rates $p$ from $\{0.01, 0.05, 0.1, 0.3\}$, with results shown in Figure 20. Additionally, we study how burn-in and the number of stored models $N$ affect the performance in SWAG. As depicted in Figures 21 and 22, we set burn-in to 60 and 120, and $N$ to 10 and 30. The results demonstrate that the Bayesian inference methods used are not highly sensitive to these hyperparameters within an appropriate range. Among these factors, the number of models $N$ has the most significant impact in SWAG, where larger $N$ has better performance, as shown in Figure 22. Across all these settings, our sequential sampling maintains similar acceleration performance.

| NFE | 1 | 3 | 5 | 8 | 12 |
|---|---|---|---|---|---|
| prior=1 | 4.3% | 3.5% | 4.0% | 3.9% | 3.9% |
| prior=10 | 1.2% | 1.5% | 2.0% | 2.0% | 2.0% |
| prior=100 | 1.0% | 1.4% | 1.7% | 1.8% | 1.8% |

Table 8: The empirical calibration error (ECE) of SGLD using sequential inference (SI) with varying Gaussian prior. We report the results using FC-DEQ on MNIST.

**Langevin Sampling.** For the Langevin step size $\tau$, we conducted experiments with SWAG using MDEQ-SMALL on CIFAR-10 by evaluating the Expected Calibration Error (ECE). The results in Table 9 indicate that our method is robust to moderate variations in $\tau$, but its performance degrades when the step size becomes excessively large since the similarities degrade.

## K   MEMEORY USAGE

In this section, we evaluate the running memory of our sequential inference to show that the memory overhead of sequential inference is marginal. The only additional memory usage arises from the classification process, where we retain the classification head and classification features beyond the peak memory usage of the backbone. For instance, the classification vector has only 200 dimensions, which is negligible compared to the input shape and fixed point representation in MDEQ-SMALL on CIFAR-10.

| NFE | 1 | 3 | 5 | 8 | 12 |
|---|---|---|---|---|---|
| $\tau = 0.001$ | 0.7% | 1.7% | 1.8% | 1.8% | 1.8% |
| $\tau = 0.005$ | 0.7% | 1.7% | 1.8% | 1.8% | 1.8% |
| $\tau = 0.01$ | 1.4% | 1.5% | 2.0% | 1.9% | 1.8% |
| $\tau = 0.05$ | 5.6% | 2.4% | 2.1% | 1.8% | 1.8% |
| $\tau = 0.1$ | 10.7% | 3.5% | 1.9% | 1.8% | 1.8% |

Table 9: The empirical calibration error (ECE) using sequential inference (SI) with varying Langevin step size $\tau$. We report the results using FC-DEQ on MNIST.

To further support this claim, we measured the memory usage of MDEQ-SMALL with a batch size of 512. The results show that the memory usage without SI is 4032 MB, while with SI it is 4244 MB, an increase of only 5.3%.

