# OpenReview forum: "Bayesian Deep Equilibrium Models with Sequential Inference"
_ICLR.cc/2026/Conference — Submitted to ICLR 2026_

### Official Review · Reviewer_fuzm · 2025-10-29

**Soundness:** 2
**Presentation:** 2
**Contribution:** 2
**Rating:** 2
**Confidence:** 4

**Summary:**

This paper extends **Deep Equilibrium Models (DEQs)** by incorporating Bayesian inference to estimate predictive uncertainty. DEQs solve a fixed‐point equation instead of using a finite sequence of layers, leading to memory efficiency and implicit depth. The authors argue that uncertainty estimation for DEQs is unexplored. They therefore construct Bayesian DEQs using Monte Carlo Markov Chain methods (Stochastic Gradient Langevin Dynamics, SGLD), SWAG and MC‐dropout. To tackle the high computational cost of repeated fixed‐point solves during ensembling, they introduce sequential inference, which has two components:

1.	Sequential sampling: a scheme to draw parameter samples from the posterior in a sequence such that consecutive samples have small $\ell_2$ distance.

2.	Sequential computation: uses the previous sample’s fixed-point solution as the initialization for the next inference. They provide a theorem (Theorem 4.1) showing that if the DEQ mapping is smooth and consecutive parameter samples are sufficiently close, the resulting fixed points are also close. This motivates reusing the previous fixed point to accelerate convergence.

The paper evaluates the proposed scheme on multiple datasets and reports that sequential inference achieves comparable accuracy and uncertainty calibration with as few as one fixed‐point iteration, yielding up to a threefold reduction in inference time. The paper further includes ablation studies investigating distance between parameter samples and fixed‐point initialization.

**Strengths:**

1.	The paper tackles an important question of uncertainty estimation in DEQs. Given the increasing use of implicit models, exploring Bayesian approaches is timely.

2.	The idea of reusing fixed‑point solutions across parameter samples is intuitively appealing and easy to implement. The theorem provides some insight into when this reuse may be valid.

3.	The authors conduct experiments on multiple datasets and provide ablations showing that sequential inference improves error, negative log‐likelihood and calibration metrics as the number of fixed‐point evaluations increases.

**Weaknesses:**

1.	**Incremental novelty:** The core ideas are largely straightforward. Warm-starting iterative solvers using previous solution is a well-known technique, and applying MCMC/SWAG to a new model class (DEQs) is incremental. The “sequential sampling” is essentially treating SWAG sample generation as a Markov chain, which has been done in other contexts (and indeed SGLD already produces correlated samples by nature). The paper does not clearly distinguish its novelty from these known practices.

2.	**Theoretical analysis is shallow:** The theorem relies on strong smoothness and proximity assumptions that may not hold, and there is no analysis of the effect on the posterior or predictive distribution. Theorem 4.1 provides a bound but under strong conditions. The assumption that all samples lie in a small smooth region is not justified for a complex model. Moreover, the paper claims sequential sampling “preserves” the posterior, but this is only briefly mentioned with a W2 distance result in App. G. There is no analysis of the bias or variance introduced by correlating samples (i.e. we lose independent samples). It is unclear how sequential sampling affects the effective sample size.

3.	**Experimental scope is limited:** Only small‐scale DEQs are considered; evaluation on ImageNet uses a small MDEQ‐SMALL model. More challenging tasks such as language modeling or video understanding are absent. There is no comparison to strong calibration baselines (e.g., temperature scaling, deep ensembles, last‐layer Laplace).

4.	**Experimental rigor:** Results are presented without error bars or multiple runs, so we cannot assess statistical significance. It is unclear if the reported speedups and performance gains are consistent across random seeds. The single-example curves suggest big differences, but more rigorous evaluation (e.g. repeated trials) would strengthen the claims.

5.	**Clarity issues:** The description of sequential sampling and its relation to posterior sampling is vague. The algorithm pseudocode mixes indices and omits loop variables. Figures lack sufficient detail and sometimes repeat content from the text.

**Questions:**

1.	**Posterior representativeness:** How does sequential sampling (especially the correlated Langevin walk for SWAG) affect the true posterior approximation? Did you measure effective sample size or posterior uncertainty calibration relative to independent samples? Under what settings (e.g. step size τ) does sequential sampling remain valid?

2.	**Baselines:** Why were MC Dropout and deep ensembles not shown in the main results? For example, how does a standard DEQ ensemble (trained with different seeds) compare in accuracy and ECE to Bayesian DEQ? Similarly, how does sequential inference apply to a DEQ ensemble?
3.	**Laplacian methods:** In Related Work you state that Laplacian approximations are not suitable for DEQs (citing Section 5.3), but I could not find the explanation in the paper. Can you clarify why LA/LLA/LLLA would not work or would be inefficient for DEQs?
4.	**Solver details:** Which fixed-point solver (e.g. Broyden’s method or simple iteration) was used in each experiment? How does the choice of solver interact with your sequential inference? For example, does Broyden’s method benefit as much from warm-start as first-order solvers?
5.	**Statistical significance:** Were the plots (e.g. Figures 5–7) averaged over multiple runs? If not, can you provide error bars or standard deviations to show the consistency of the speedup?
6.	**Sequential sampling validity:** Can you provide a theoretical guarantee that the proposed Langevin‐based sequential sampling preserves the correct posterior distribution? How sensitive are the results to the Langevin step size τ (for SWAG sampling) and the number of burn-in/skip steps (for SGLD)? Table 9 hints at τ-dependence, but more discussion is needed on how one should choose these parameters.
7.	**Scalability:** Can you demonstrate the method on larger DEQ architectures (e.g., multi‐scale or vision transformer–based DEQs) and more challenging datasets? What is the overhead when the base DEQ is large?

**Details Of Ethics Concerns:**

I did not identify explicit violations of the ICLR code of ethics.

---

### Official Review · Reviewer_NEKE · 2025-10-30

**Soundness:** 3
**Presentation:** 3
**Contribution:** 3
**Rating:** 6
**Confidence:** 4

**Summary:**

This paper proposes Bayesian Deep Equilibrium Models (Bayesian DEQs) that integrate Bayesian uncertainty estimation into implicit equilibrium networks. To address the high computational cost of Bayesian inference in DEQs, a Sequential Inference mechanism is also proposed to reuse information across posterior samples through sequential sampling and computation. Experiments on MNIST, CIFAR-10, and ImageNet show that this approach achieves up to 3× faster inference without sacrificing predictive accuracy or uncertainty quality.

**Strengths:**

1. The problem formulation is novel. The paper addresses an unexplored but important challenge of uncertainty estimation for DEQs, which has good practical values.

2. The proposed Sequential Inference framework achieves notable computational acceleration for Bayesian DEQs without sacrificing performance.

3. The paper provides theoretical justification through bounded equilibrium analysis and extensive empirical evaluations on multiple benchmarks, demonstrating its robustness and generalisation.

**Weaknesses:**

1. There is a lack of quantitative evaluation on how sequential dependence affects the posterior fidelity, i.e., whether the sampled sequence still faithfully represents the true Bayesian posterior. Metrics such as effective sample size, autocorrelation, or divergence from standard MCMC samples are not reported, leaving it unclear whether the sequential samples still faithfully capture the underlying posterior distribution.

2. The sequential computation step reuses previous fixed points for initialization. Although Theorem 4.1 offers a local boundedness guarantee under smoothness assumptions, there is no global or empirical analysis of whether this reuse affects convergence stability or the uniqueness of equilibrium solutions, particularly under large parameter perturbations.

**Questions:**

In scenarios where the DEQ mapping admits multiple equilibria, how does reusing a previous fixed point influence which equilibrium the solver converges to?

---

### Official Review · Reviewer_Dajx · 2025-11-01

**Soundness:** 3
**Presentation:** 1
**Contribution:** 2
**Rating:** 4
**Confidence:** 3

**Summary:**

This paper addresses the high computational cost of performing Bayesian inference in Deep Equilibrium Models (DEQs), where each posterior sample requires solving an implicit fixed point. Authors propose Sequential Inference, which exploits the smooth evolution of parameters in methods like SGLD and SWAG to reuse the previous fixed point as initialization for the next DEQ solve. A short theoretical result shows that fixed points vary smoothly with small parameter changes, justifying the warm-start strategy. Experiments on MNIST, CIFAR-10, and ImageNet show up to 3× faster inference with similar calibration and accuracy.

**Strengths:**

- The proposed sequential inference idea is simple, effective, and can be integrated into existing SGLD or SWAG pipelines with minimal changes.

- The theoretical result (Theorem 4.1) correctly formalizes the intuition that small parameter perturbations induce small changes in the DEQ fixed point. Although interesting, it's not too surprising.

- Experimental evidence across a few datasets shows consistent speedups without degrading predictive uncertainty calibration.

**Weaknesses:**

- The narrative is difficult to follow: definitions of DEQs and standard Bayesian methods are scattered through the methodology section rather than being grouped as background, making it hard to discern what is new. The paper would benefit from a dedicated background section clearly distinguishing DEQs from implicit layers and neural differential equations, as well as summarizing existing UQ methods.

- The method essentially combines (i) correlated posterior sampling (which SGLD and SWAG already produce) with (ii) warm-starting the DEQ solver. This is more of a practical engineering optimization than a fundamentally new Bayesian principle.

- Theorem 4.1 only provides a qualitative bound on the proximity of fixed points; it does not predict the actual reduction in the number of solver iterations. Although not necessarily a problem, it is worth noting.

- The paper modifies the sampling process to make samples more correlated, but does not analyze how this affects posterior diversity or effective sample size. Authors could look into metrics like ESS or variance decomposition.

- The method is demonstrated only for SGLD and SWAG on vision DEQs. No results are shown for other types of DEQs such as implicit GNNs or PDE solvers, despite the generality claims.

**Questions:**

1. Could the authors clarify how much of the speedup arises from sample correlation versus solver warm-starting?
2. How does the method behave when posterior samples are farther apart, or when the DEQ map has multiple fixed points?
3. Are there failure cases where reusing a previous fixed point leads to convergence issues?
4. Could sequential inference be applied to other implicit models such as Neural ODEs and what would be the limitations? I am still not 100% of the difference.
5. What is the impact on posterior mixing and effective sample size when explicitly correlating SWAG samples using the Langevin noise update?

---

### Official Review · Reviewer_CGof · 2025-11-01

**Soundness:** 2
**Presentation:** 1
**Contribution:** 2
**Rating:** 2
**Confidence:** 3

**Summary:**

The paper proposes Bayesian version of Deep Equilibrium Models (DEQs) based on the Stochastic Weight Averaging-Gaussian (SWAG) using two techniques: sequential sampling and sequential computation. Benchmark tests show 3 times speed up compared with standard SWAG algorithm.

**Strengths:**

It is the first attempt to address the inference of DEQ. The proposed sequential procedure can capture the similarity of parameters and use it to reduce redundant computation.

**Weaknesses:**

The presentation is not clear. It only clarifies that the similarity is NOT correlation in the Monte Carlo but does not define what it is. The notations are confusing and the implication of the main theorem is unclear. What is the advantage in terms of error rate if that is the same as SGD?

**Questions:**

1. What is the similarity of parameters? Do you refer to their distance being small? Please clarify.

2. What is the difference between $z^*_i$ and $z^*$ in the statement of Theorem 4.1? What does the inequality imply? Can $\delta$ here be arbitrarily small? If not, how does it demonstrate the closeness to the fixed-point solution?

3. Figure 3 is so confusing. Please consider improve the explanation in the main text.

4. The error rate of the proposed method is the same as SGD across different figures. Is it expected? What is the advantage in terms of this?

Minor things:

Line 061: spell out MDEQ when mentioning for the first time.
Line 286: typo 'compatation'.

---

### Meta-Review · Area_Chair_aEuR · 2025-12-29

**Summary:**

This work builds on top of the deep equilibrium models (DEQs) and develops a Bayesian version of them. Key to the paper's contribution is to speed-up approximate Bayesian inference methods (Langevin sampling and SWAG) when applied to DEQs. The idea is to reduce the redundancy of similar forward process (by similar network weights), i.e., using the fixed point (DEQ's solver solution) from the previous weight-updating iterations as the initialisation of the fixed point for the DEQ with current weight. Theoretical analysis provides an error bound for this "warm-start" initialisation method. Experiments on synthetic and real-world classification datasets (Cifar-10 and ImageNet) + regression datasets (UCI Energy) demonstrate the effectiveness of the proposed approach.

After reading the reviewers' concerns (see below box) and given that no author rebuttal is provided, I made the decision of rejection at this time. From my brief read of the submission, I would give the following suggestions to the authors in future submissions:
- Better positioning of the narrative. I personally think the paper has done a good job of demonstrating many existing BNN approaches (including those in appendix) on DEQs, so the authors might consider a narrative based on **approximating posterior with fast solutions**.
- Consider generalisation of the approach. Other networks beyond DEQs that also require fixed points or solvers (e.g., Bayesian version of neural ODEs) might benefit from the proposed approach.
- Comparison to non-equilibrium based approaches: as the experiments are based on regression and classification, how would Bayesian DEQs compare with normal BNNs?

**Reviewer Concerns:**

Reviewers' major concerns include:
- Clarity issues of the presentation.
- Theoretical analysis is interesting but not deep enough: little discussion on whether the proposed sequential inference method can sample from the exact posterior, and if not how big the error is.
- Experimental evaluations do not provide metrics related to posterior sampling quality e.g., ESS.

Some reviewers are concerned regarding the lack of other approximate Bayesian inference methods, but more results on them are included in appendix.

No author rebuttal is provided.

**Reviewer Scores:**

The majority of reviewers voted for rejection, except for one reviewer (marginally above the acceptance threshold. But would not mind if paper is rejected), who also questioned about some of the quantitative results.

No author rebuttal is provided.

---

### Decision · Program_Chairs · 2026-01-26

Reject